# Seroprevalence and Molecular Characterization of *Leptospira* spp. in Rats Captured near Pig Farms in Colombia

**DOI:** 10.3390/ijerph191811539

**Published:** 2022-09-14

**Authors:** Sara López-Osorio, Diego A. Molano, Anderson López-Arias, Nélida Rodríguez-Osorio, Corina Zambrano, Jenny J. Chaparro-Gutiérrez

**Affiliations:** 1Grupo de Investigación CIBAV, Facultad de Ciencias Agrarias, Universidad de Antioquia-UdeA, Medellín 050034, Colombia; 2Unidad de Genómica y Bioinformática, Departamento de Ciencias Biológicas, Centro Universitario Regional Litoral Norte, Universidad de la República, Rivera 1350, Salto 50000, Uruguay; 3Asociación Porkcolombia-FNP, Ceniporcino, Bogotá 111311, Colombia

**Keywords:** synanthropic rats, microagglutination tests, phylogenetic analysis

## Abstract

Gram-negative spirochete *Leptospira* spp. causes leptospirosis. Leptospirosis is still a neglected disease, even though it can cause potentially fatal infections in a variety of species including humans. The purpose of this study was to determine the seroprevalence of leptospirosis in pig farm captured rodents and characterize the isolated samples. Rats were captured, sampled, and euthanized in the vicinity of pig farms to obtain serum for microagglutination tests (MAT) and kidney tissues for PCR amplification of the 16S rRNA and *LipL32* genes. A fraction of the 16S rRNA PCR product was sequenced and phylogenetically analyzed. The results showed a *Leptospira* seroprevalence of 13.8% (77/555) among the 555 captured rats. PCR positivity for *Leptospira* spp. reached 31.2% (156/500), and the positivity for pathogenic *Leptospira* spp. was 4% (22/500). Phylogenetic analysis matched eight samples with *L. interrogans* serovar icterohaemorrhagiae and two with *L. interrogans* serovar pyrogenes. Two sequences were located within the pathogenic *Leptospira* clade but did not match with any specific strain. The seroprevalence found in the rats around swine farms indicates a potential risk of transmission to the pigs. The identification of pathogenic *Leptospira* outlines the importance of more research as well as updating the current strategies for the diagnosis, control, and prevention of porcine leptospirosis in Colombia.

## 1. Introduction

Pathogenic *Leptospira* produce a zoonotic disease known as leptospirosis. Over one million human leptospirosis cases, with approximately 60,000 deaths, are predicted to occur globally each year [1]. As a result, leptospirosis has become one of the most widespread zoonotic illnesses on the planet [2,3]. Leptospirosis has been traditionally linked to Third-world nations, precarious situations, and agriculture practices. However, recent research suggests that even metropolitan areas in developed countries can have emerging leptospirosis cases, which are more common during times of heavy rainfall and flooding [4], or after natural disasters such as hurricanes, which can trigger disease outbreaks [5]. Despite Colombia’s diverse environment, a significant portion of the country has a tropical climate with the potential for constant rainfall and flooding due to the El Niño Southern Oscillation [6,7]. As a result, the Colombian territory is vulnerable to a high incidence of leptospirosis [8].

A significant part of Colombia’s territories is used for agricultural purposes. There are approximately 440,156 square miles of surface area in Colombia. Of these, 19,300 square miles are used for agriculture (4%), and over 131,274 square miles (30%) are used for livestock [9]. Pig farming is one of Colombia’s most important agricultural sectors, and is rapidly expanding. In 2019, nearly five million pigs were slaughtered for national consumption, and this number is expected to rise [10]. Because of the economic losses caused by reproductive decline due to leptospirosis, this disease may have a significant effect on livestock [11].

The main reproductive issues caused by leptospirosis in swine farms include abortions, occurring 2–4 weeks before term, early-onset parturition, where a premature delivery leads to the birth of weak or dead piglets, and empty sows at farrowing. Piglets born to infected sows become weak, anemic, and jaundiced, usually resulting in their death soon after birth [12]. Infected lactating sows may be lethargic, develop anemia and jaundice, along with fever, which left untreated, may cause the death of the animal [13].

Leptospirosis is spread through direct contact with the infected animals’ urine or kidney as well as indirectly through a contaminated environmental resource such as water and soil [14]. Rodents are known to bear a variety of pathogenic *Leptospira* spp., promoting disease transmission in humans and animals [15]. Studying the prevalence of leptospirosis in rodents is a good early spatial indicative of leptospirosis in an environment [16]. Rodents can also be used as sentinels to detect evolving *Leptospira* pathogenic strains [17]. Studies suggest that being able to diagnose and adequately control the reservoir animals of a zoonotic disease is an effective method for managing outbreaks [18]. Having knowledge of circulating pathogenic *Leptospira* spp. and its serovars in a reservoir species is beneficial mainly because it allows for a more efficient diagnosis, prevention, and control of the disease [19].

The aim of this study was to determine the seroprevalence of leptospirosis in rats captured near pig farms in Colombia using a microscopic agglutination test (MAT) and molecular identification of *Leptospira* using amplification of the genes 16S rRNA and *LipL32* via the PCR of renal tissue. For better identification of *Leptospira* serovars, 16S rRNA gene amplicons were sequenced and analyzed to infer the phylogenetic relations of the identified serovars.

## 2. Materials and Methods

### 2.1. Location of Sampling

Rats were captured between June and December 2015. The study’s sampling sites were selected from among the members of Porkcolombia, the country’s largest pig farmer association. Sixty-three farms were chosen as capturing sites out of a total of 1066 possible Porkcolombia-associated pig farms. The geographic region of the selected farms includes 65.80% of the total pig farming area in Colombia. The chosen farms were located in the departments of Antioquia, Valle del Cauca, Cundinamarca, Meta, and three departments from the Coffee-Growing Axis: Caldas, Quindío, and Risaralda.

### 2.2. Rats Collecting and Sampling

Rats were caught in traps (23 cm × 10 cm × 14 cm) using bacon, with garlic salt added, as bait. Traps were strategically placed near rat nests or places where rats are known to eat. Corrals, pig feeders, rooftops, warehouses, and hallways were also included. Traps were monitored every four hours for a total of five nights. Only synanthropic rats were included in the study.

Individuals captured were categorized using the morphological metric key described by Elias [20]. *Rattus rattus* was identified as having a pointed nose, large eyes and ears, a light slender body, and a tail longer than the head–body length. *Rattus norvegicus* has a blunt nose, small eyes and ears, a heavy thick body, and a tail shorter than the head–body segment, as described by Brown [21]. Characteristics such as sex, age group, and place of capture were reordered for every individual. Rats were given a dose of ketamine at 50–60 mg/kg applied intramuscularly. Once the rats underwent anesthesia, an intracardiac blood sample of 3 mL was taken and placed in a tube without an anticoagulant. Euthanasia was performed by an intracardiac injection of sodium phenobarbital at 200 mg/kg. A full necropsy was performed on all captured rats using new sterile material for each animal to avoid cross-contamination. Kidneys were taken from each animal for DNA isolation and molecular characterization. Carcasses were frozen at −80 °C and disposal was achieved by incineration.

### 2.3. Microagglutination Test for Leptospirosis (MAT)

Serum samples were centrifuged at 1500 rpm for 8 min and blood serum was stored for MAT to identify *Leptospira* serovars, positive samples as proposed by the WHO [22]. The test uses live cultivated antigens in a liquid medium, Ellighausen–McCollough–Johnson–Harris, supplemented with *Leptospira* complemented serum.

Eight pathogenic serovars of *Leptospira* were analyzed: pomona, canícola, icterohaemorrhagiae, grippotyphosa (*L. interrogans*), ballum, tarassovi (*L. borgpeterseneii*) Bratislava, and autumnalis (*L. kirschneri*). Initially, dilutions of 1:50–1:200 were prepared and added to 50 μL of the *Leptospira sp*. antigen. Then, the samples were incubated for 2 h at 28–30 °C. For reading, one drop, 5 μL, was taken from each trap and placed on a slide for observation at 10× under a dark-field, Olympus^®^ CX41 microscope. Positive serum samples were those that had a dilution greater or equal to 1:100, along with a 50% agglutination of Leptospires. Positive samples were placed in dilutions from 1:100 up to 1:3200 to determine the antibody titers. The test was standardized in the Special Parasitology Lab at the Diagnostic Unit of the Faculty of Agrarian Sciences, Universidad de Antioquia. Strains were donated by the microbiology laboratory of the Universidad de Córdoba (Colombia).

### 2.4. Data Analysis

Data were processed using Microsoft Excel© version 16.17 (Albuquerque, NM, USA) for descriptive statistics, central tendency, and dispersion measures, and the constructions of means, tables, and graphs. In order to obtain an epidemiologic prevalence of *Leptospira* spp. in rats, the following formula was used.
$$p=\frac{\mathrm{Number}\;\mathrm{of}\;\mathrm{seropositive}\;\mathrm{rats}\;\mathrm{captured}}{\mathrm{total}\;\mathrm{number}\;\mathrm{of}\;\mathrm{rats}\;\mathrm{captured}}\times 100$$

Serology data were submitted to a descriptive analysis based on the frequency and percentage of qualitative variables with a confidence interval of 95%. The association between seroprevalence and other variables such as sex, species, and age of the rats were analyzed via a Chi-square test of independence with a value of *p* < 0.05. A 2 × 2 contingency table was used for the comparison of all variables. Frequencies were calculated to determine the variable independence. The odd ratio (OR) was determined for all category type data to evaluate the probability of *Leptospira* spp. seropositivity. Statistical analysis was performed in Microsoft Excel^®^ 2010 (Albuquerque, NM, USA) and SPSS^®^ software IBM Corp., 2017 (Armonk, NY, USA).

### 2.5. Polymerase Chain Reaction Test (PCR)

Five hundred kidney samples were subjected to PCR amplification using two sets of primers. *Leptospira* spp., specific primers for gene 16S rRNA amplify a 330 bp fragment, and primers for the gene *LipL32*, specific for pathogenic *Leptospira* [23], which amplify a 660 pb fragment. The reaction mix included 400 mM of dNTP, 1 mM of each primer, 4 mM of ${\mathrm{MgCl}}_{2}$, 1× of PCR buffer solution, 2.5 U of *Taq* polymerase, Invitrogen^®^, and 1 μL of DNA (40 ng/μL). The total volume of the reaction medium was 25 μL. The amplification conditions were as follows: denaturation at 95 °C for 3 min, followed by 30 cycles of 95 °C for 30 s, then 58 °C for 30 s, next at 72 °C for 40 s, and a final extension at 72 °C for 5 min.

A second round of PCR was run on 50 positive *Leptospira* samples. A 330 bp fragment of 330 of gene 16S rRNA was amplified using primers F 5′-GGCGGCGCGTCTTAAACATG-3′ and R 5′-TTCCCCCCATTGAGCAAGATT-3′ [24]. The reaction was carried out in a volume of 50 μL using 10× Taq buffer, 3 mM of ${\mathrm{MgCl}}_{2}$, 200 μM of dNTP, 0.5 U of *Taq* polymerase, and a pair of primers (each 10 μM) with 50 ηg of DNA. Amplification conditions were an initial denaturation at 95 °C for 5 min, and then 30 cycles of denaturation at 94 °C for 1 min, annealing at 60 °C for 45 s, an extension at 72 °C for 1 min, followed by a final extension at 72 °C for 7 min. The PCR product was purified and sent to the Macrogen^®^ Company, Seoul, Korea for Sanger sequencing.

### 2.6. Sequence Assembly

Contig sequences were assembled using forward and reverse sequences. The minimal overlap length was set at 10 bp. Nucleotide differences between the forward and reverse sequences were resolved based on the sequence quality scores. The assembled contigs were subjected to quality control using the QC module of the CLC Genomics workbench 20.0.4 software (https://digitalinsights.qiagen.com/). Contig sequences were used if they passed all of the quality filters. Only 12 out of the 50 samples passed the contig sequence quality filters. For 14 of the samples, it was not possible to build a contig sequence; although the reverse sequences were of good quality, the forward sequence failed the quality check. Therefore, only the reverse sequence could be used for an exploratory alignment. However, only samples with a contig sequence were used for phylogenetic tree construction and analysis. For the remaining samples, either the forward, reverse, or both sequences failed the quality check and were completely excluded from the analyses.

### 2.7. Phylogenetic Analysis

For the phylogenetic analysis, sequences were aligned using the *CLC Genomics Workbench* 20.0.4 alignment module (https://digitalinsights.qiagen.com/). Sample sequences were aligned with 46 sequences corresponding to different *Leptospira* species and serovars. Of these, 11 different pathogenic, five intermediate pathogenic, and seven nonpathogenic *Leptospira* species, along with different serovar sequences, were used. The pathogenic strains used were *L. alexanderi manzhuang*, *L. alexanderi manhao*, *L. mayottensis*, *L. weilii calledoni 1*, *L. weilii calledoni 2*, *L. weilii topaz*, *L. borgpetersenii javanica 1*, *L. borgpetersenii javanica 2*, *L. interrogans 1*, *L. interrogans 2*, *L. interrogans icterohaemorrhagiae 1*, *L. interrogans icterohaemorrhagiae 2*, *L. interrogans pyrogenes 1*, *L. interrogans pyrogenes 2*, *L. interrogans pyrogenes 3*, *L. interrogans pyrogenes 4*, *L. interrogans pyrogenes 5*, *L. interrogans lai*, *L. kirschneri grippotyphosa*, *L. kirschneri cynopteri*, *L. noguchii panama*, *L. alstonii pingchang*, *L. kemtyi malaysia*, *L. santarosai kobbe*, and *L. santarosai shermani.* The intermediate strains used were *L. fainei hurstbridge 1*, *L. fainei hurstbridge 2*, *L. inadai lyme*, *L. inadai arguaruna*, *L. broomii 1*, *L. broomii 2*, *L. wolffi khorat*, and *L. licerasiae varilla*. The nonpathogenic strains used were *L. yanagawae saopaulo 1, L. yanagawae saopaulo 2, L. meryeri ranarum, L. Terpstrae hualin 1,*
*L. terpstrae hualin 2, L. wolbachii codice, L. biflexa patoc 1, L. bifelxa patoc 2*, *L. vanthielii holland 1*, *L. vanthielii holland 2*, and *L. idonii 1*, *L. idonii 2.* For the alignment, a penalty of 10 was set for the gap opening and a penalty of 2.0 for the gap extension. CLC Genomics Workbench 20.0.4 (https://digitalinsights.qiagen.com/) (QIAGEN, Aarhus C, Denmark) was also used for the model selection and maximum-likelihood tree inference. The selected model was Kimura80 with a transition/transversion value of 2.0. To evaluate the tree topology, 1000 Bootstrap reiterations were performed. Nodes with a Bootstrap value below 50 were collapsed.

### 2.8. Ethics Statement

The capture, handling, euthanasia, and sampling of animals was approved by Act 99 of 29 September 2015. Samples were collected and processed under an agreement between the Universidad de Antioquia and the Asociación Porkcolombia-FNP. The project was funded by these entities and the Ministerio de Agricultura y Desarrollo Rural (Agreement 15 November 2015). Access to private property for sample collection was granted by the owners.

## 3. Results

In total, 555 rats were collected across 66 pig farms from seven departments in Colombia. Most rats were captured from pig farms in the departments of Antioquia, 333 (60%, IC95: 55.79–64.10), and Cundinamarca, with 142 (25.5%, IC95: 22–29.42). Smaller numbers of rats were captured in the department of Valle del Cauca, 34 (6%, IC95: 4.2–8.4) and the Coffee-Growing Axis with 31 (5.5%, IC95: 3.8–7.8). Only 15 rats (3.3%, IC95: 1.52–4.42) were captured in the department of Meta.

Most rats were captured within the farms in warehouses, feeding places, trash piles, and near rodent nests. Captured rats belonged to the family Muridae, subfamily Murinae, genus *Rattus*, and two species: *Rattus norvegicus* and *Rattus rattus*. Seventy-four (410/555, CI95: 70.35–77.65) percent of captured rodents belonged to the species *Rattus norvegicus* and 26% (145/555, CI95: 22.35–29.65) to the species *Rattus rattus*. Among the captured animals, 320 were females (320/555, 58%, IC95: 53.43–61.81) and 235 males (235/555, 42%, IC95: 38.19–46.57). Seventy eight percent of the captured rats were adults (434/555, CI95: 74.55–81.41) and the remaining 22% (121/555, CI95:18.55–25.45) were juveniles. The sex and age distribution of the captured rats is recorded in Table 1.

Seropositivity to *Leptospira* spp. through MAT was found in rats captured from 39 farms (39/66, 61.9%, IC95: 48.8–73.86). Only 13.8% (77/555, IC95: 11.11–16.89) of the captured rats presented antibodies against *Leptospira* spp. The highest ratio of seropositive rats was found in Valle del Cauca with six seropositive rats out of 34 (17.64%, IC95: 6.73–34.53). In Cundinamarca, 22 rats were seropositive out of 142 (15.49%, IC95: 9.97–22.51). In Antioquia, 45 rats were seropositive out of 333 captured (13.51%, IC95: 10.03–17.66). In the Coffee-Growing Axis, only three out of 31 captured rats were seropositive to *Leptospira* (9.6%, IC95: 2–25.75). In Meta, only one rat was seropositive out of the 15 captured there. A layout of the distribution of rats captured by department and their seropositivity status can be seen in Table 2.

The most common serovar found was *Leptospira interrogans*, serovar *icterohaemorrhagiae*, with 21 samples (21/77:27.27%, IC95: 17.74–38.62). Serovar canicola was found in 12 samples (12/77, 15.58%, IC95: 8.32–25.64). Serovar tarassovi and Pomona were each found in nine samples (11.68%, IC95: 5.49–21.03). Serovar grippothyphosa was found in eight samples (8/77, 10.38%, IC95: 4.59–19.45). Serovar ballum was found in six samples (6/77, 7.79%, IC95: 2.91–16.19) and serovar autumnalis was found in five samples (5/77, 6.49%, IC95: 2.14–14.51). The distribution of the *Leptospira* serovars by department is presented in Table 3.

The species *Rattus norvegicus* presented a seroprevalence of 11.9% (49/410, IC95: 8.8–15.1), while in *Rattus rattus*, the seroprevalence was 19.31% (28/145, IC95: 12.9–5.7). A disproportional seroprevalence was found for *R. norvegicus* 63% (49/77, 52.9–74.4). No differences were observed for juvenile vs. adults or for males vs. females (chi-sq., *p*’s > 0.05).

The PCR results showed that 69% (344/500) of the kidney samples were negative for *Leptospira* spp. and 31.2% (156/500, IC95: 27.16–35.46) were positive. From these, 26.8% (134/500 IC95: 22.96–30.91) corresponded to *Leptospira* (with only 16S RNA gene amplified) and 4% were positive for pathogenic *Leptospira* (22/500, IC95: 2.78–6.59) (with *LipL32* gene amplified).

The phylogenetic tree inferred by maximum likelihood after the alignment of 16S partial sequences showed that 8/12 (66.66%) samples matched the pathogenic *L. interrogans* serovar icterohaemorrhagiae, 2/12 (16.66%) samples matched the pathogenic *L. interrogans* serovar pyrogenes, and 2/12 (16.66%) samples did not match any specific *Leptospira* strain, but were located within the pathogenic *Leptospira* clade. Figure 1 shows the phylogenetic tree made using the maximum likelihood method. Within the figure, it can be observed that 40_LEP-F Consensus-RC and 254_LEP-F Consensus did not align with any specific strain.

## 4. Discussion

The findings revealed that rats were abundant in most pig farms as well as a high seroprevalence of *Leptospira* in these reservoirs. Some farms had low rodent capture rates. Cats roamed freely on the premises of these farms in large numbers. This occurrence was particularly noticeable in several farms in Antioquia. The low capture rates of rats in these cases can be attributed to the presence of predatory domestic animals such as cats and dogs, which are thought to scare rats away [26]. Discovering rodent nests aided in the capture of specimens in these situations.

Most of the farms studied use hog flooring for young piglets. Subterranean tunnels connect the flooring to the drainage system, which leads to a common drain. As a result, rats could freely wander in the subterranean system and return to their nests without being detected. Other pig pens had cement floors and a hole for drainage. During the night, the rats had the possibility of passing through the drainage hole and entering and exiting the pig pens. This was evidenced while capturing the rats. In various farms where *Rattus rattus* were captured, individuals came down from the rooftops through water tubes or columns of the pig housing. The infrastructure and activity found in the farms allowed for a persistent presence of rats in and around the facilities, potentially enabling the infectious disease transmission of *Leptospira* spp., being one of the most important pathogens that could potentially be transmitted under these conditions [27].

When looking at the seropositivity found by region, the department of Valle del Cauca had the highest value of 17.64% (6/34), and Meta had the lowest value of 6.66% (1/15). These findings can be attributed to Valle de Cauca’s mean annual temperature of 22 °C, which has previously been shown to be an optimal temperature for peak prevalence [28]. The prevalence of *Leptospira* significantly drops the closer it gets to 30 °C. This can partially explain why Meta had the lowest prevalence, since it had the highest annual mean temperature, 26 °C. Regarding precipitation, El Valle del Cauca had the highest precipitation, which, as mentioned before, is correlated with a high prevalence of *Leptospira* [4]. The mean altitude of the regions showed no correlation with positivity, which matches the finding in Biscornet et al. 2021 [29].

The present study identified 13.8% (77/555, IC95: 11.11–16.89) *Leptospira* seropositivity against eight serovars in rodents captured on pig farms. The presence of *Leptospira* found in rats is important because it identifies a risk factor for transmission to domestic animals and humans [30,31,32]. Giraldo de Leon et al. [33] found that 1/30 (3.3%) of rats captured on pig farms in Colombia’s Coffee-Growing Axis were seropositive to *Leptospira* spp. by MAT. That seroprevalence was lower than the one observed in this study for that region, which was 9.6% (3/31). The difference in seroprevalence between reports can be caused by various factors. The present study took place during months of higher rainfall [34]; a possible explanation for a higher seroprevalence in the present study compared to that of Giraldo de Leon et al. [33] could be that rats have a higher seroprevalence of *Leptospira* spp. during higher rainfall periods, which could affect transmission, the rat population dynamics, rat behavior, etc. However, more studies need to be conducted to confirm these types of associations. Another explanation for the difference in seroprevalence between the two studies is that different serovars were used for the MAT. *Leptospira interrogans* serovars Ballum, tarasovi., and autumnalis were not included in Giraldo de Leon’s research. These serovars make up 20 out of the 77 (30%) seropositive results in this study. This indicates that these serovars are prevalent in the Colombian territory and the study by Giraldo de Leon potentially could have yielded a higher seroprevalence were they included in the study. Additionally, this shows the importance of promoting the use of molecular and isolation tests as support methods when conducting serodiagnosis, hence, allowing one to correctly see the state of infection in a population, which may be higher than that which is reported [32].

A systematic review of the seroprevalence of *Leptospira* in Colombia, published by Carreño et al. [8], found that pigs in Colombia had a 12% seroprevalence for *Leptospira* spp. Even though the present study was not aimed directly at seropositivity in pigs, it focused on one synantropic animal to carry out the bacteria in an area. There seems to be a possible correlation between the seroprevalence of *Leptospira* spp. in the rats in this study (13.8%) and the seroprevalence of *Leptospira* spp. in pigs (12%) in Colombia.

Boey et al. [15] found that, in a variety of *Leptospira* spp. seroprevalence studied in rats worldwide, the species *R*. *norvegicus* has a higher seroprevalence than *R. rattus*. This statement is compatible with the results in the present study. The findings suggest that rodent species may have correlations with susceptibility to *Leptospira* infection.

The diagnosis of *Leptospira* spp. in rats is confirmed via molecular testing (PCR) in DNA isolated from the kidney or urine. This was performed to detect individuals that do not have antibody titers but are possibly disseminating the microorganism. Most articles where PCR was performed to find the prevalence of *Leptospira* in rats yielded a higher prevalence than those that used MAT [15]. MAT only measures the predetermined serovars of *Leptospira* while PCR can test for any serovar, however, the current methods can only identify this at the species level [15]. This can explain why there was a higher positive percent of *Leptospira* spp. found in the PCR test (31.2%, 156/500) in this study in comparison with the positive results established through MAT (13.8%, 77/555). Finally, an animal with an acute infection can have *Leptospira* spp. in the renal tissue, but may not yet produce antibodies, also supporting the higher PCR over MAT results.

Recent genetic characterization of *Leptospira* spp. allows for the classification of the bacteria into four subclades, S1 and S2 being saprophytes, and P1 being pathogens and P2 being intermediate [35]. It is essential to characterize the species and serovars of circulating *Leptospira* in a location to adequately implement control strategies [36]. The high cost of maintenance and genetic analysis has limited this type of research in endemic countries [37]. Limited information has been reported on the specific molecular identification of *Leptospira* spp. regarding rats in Colombia. Romero-Vivas et al. [36] found that all of the sampled rats and pigs for pathogenic *Leptospira* spp. in Colombia were compatible with *Leptospira interrogans*. Another genetic characterization study conducted in Colombia by Peláez-Sanchez et al. [38] also reported *Leptospira interrogans* as the species of *Leptospira* found in rats. The previously mentioned articles and this study differed in the methods used to sequence and identify *Leptospira* spp. and thus the findings are not completely comparable. The two previously mentioned articles only took samples from few numbers of rodents, thereby not giving a broad genetic characterization. Additionally, the rodents were not captured in pig farms but rather from a more urban location.

The phylogenetic analysis grouped all the samples within the pathogenic leptospirosis clade. There were two samples that did not match any of the reference sequences. Broader studies with more reference serovars are needed to verify the diversity of *Leptospira* spp., thereby potentially reporting new serovars if the sampled serovars do not match any of the reference serovars.

There were consensus sequences that matched the *L. interrogans* serovars pyrogenes and icterohaemorrhagiae. *L. interrogan* icterohemorrhagica is one of the most virulent serovars among the pathogenic species of *Leptospira* [39]. This serovar is known to cause mild to severe infection in humans and swine. The pyrogenes serovar has been previously found in rats in Colombia [40]. However, it does not seem to be relevant in pigs [12]. Further studies would be useful for a better identification of the present serovars to correctly prevent leptospirosis.

## 5. Conclusions

This study found a seroprevalence of 13.8% (77/555) in rats captured within pig farms in Colombia. Valle del Cauca had the highest seroprevalence, which was associated with its optimal annual mean temperature of 22 °C, and the highest precipitation among the regions in the study. Meanwhile, the hottest department, Meta, with an average mean annual temperature of 26 °C, had the lowest seroprevalence. The amplification of a segment of 16S rRNA and *LipL32* genes of the rodent renal tissue yielded a 31.2% (156/500) positive result for *Leptospira* spp. and 4% positive result for pathogenic *Leptospira* spp. 16S rRNA based phylogenetic analysis matched 66.66% of the samples (8/12) with *Leptospira interrogans* serovar icterohaemorrhagiae and 16.66% (2/12) with *Leptospira interrogans* serovar pyrogenes, another 16.66% (2/12) did not match any species and serovar of *Leptospira*, but was found within the pathogenic clade. The seroprevalence established in rats in this article is comparable with the seroprevalence previously reported in pigs in Colombia, indicating a possible correlation between the two. Moving forward, it would be beneficial to have more studies further investigate the phylogenetic analysis of circulating *L. interrogans* species and serovars. This should be followed by the updated diagnostic methods, control, and prevention strategies of Leptospirosis in Colombia.

## Figures and Tables

**Figure 1 ijerph-19-11539-f001:**
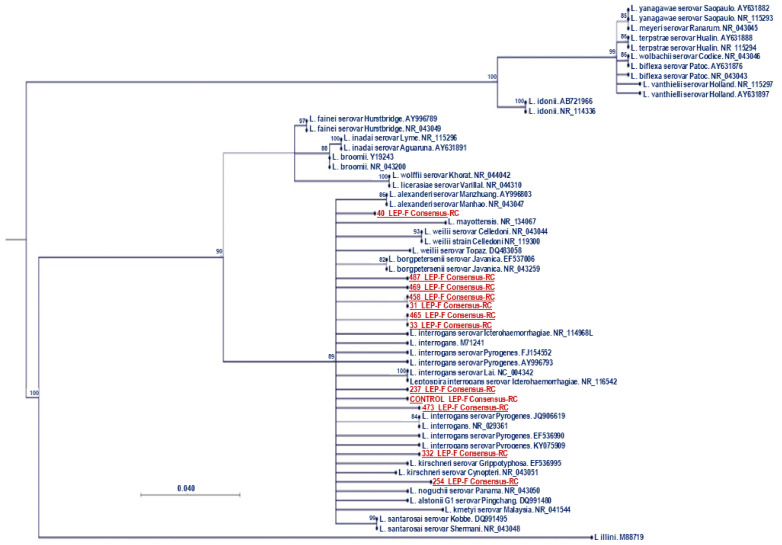
Maximum likelihood phylogenomic tree inferred from partial 16S rRNA sequences from 12 sample sequences (in red) and 45 *Leptospira* sequences from GenBank (in dark blue). Node support Boostrap values above 80 are indicated at the nodes.

**Table 1 ijerph-19-11539-t001:** The layout of the species, sex, and age group of rats captured on the pig farms in Colombia.

Species	Percent	Sex	Age Group
*Rattus norvegicus*	73.87% (410/555)	♂ 179	J 16
			A 163
		♀ 231	J 50
			A 181
*Rattus rattus*	26.13% (145/555)	♂ 53	J 19
			A 34
		♀ 92	J 36
			A 56
Total	100%	555	555

♂: Males; ♀; Females; J: juvenile; A: adults.

**Table 2 ijerph-19-11539-t002:** The layout of the temperature, precipitation, altitude, number of rats captured, and percent of seropositive rats for *Leptospira* spp. found in pig farms by region.

Department	Annual Mean Temperature	Annual Precipitation	Mean Altitude	Number of Rats Captured Per Department	Percent of Seropositive Animals
Antioquia	23 °C	3500 * mm	2099 ** mamsl	333 (60%)	13.51% (45/333)
Cundinamarca	20 °C	2500 * mm	3341 ** mamsl	142 (25%)	15.49% (22/142)
Valle del Cauca	22 °C	4000 * mm	1571 ** mamsl	34 (6%)	17.64% (6/34)
Eje Cafetero	20 °C	2500 * mm	2745 ** mamsl	31 (6%)	9.57% (3/31)
Meta	26 °C	3000 * mm	2100 ** mamsl	15 (3%)	6.66% (1/15)
Total	-	-	-	555	13.8% (77/555)

***** mm: milimeters. ** mamsl: meters above mean sea level. Data for the temperature, precipitation, and altitude were taken from the Institute of Hydrology, Meteorology, and Environmental Studies [25].

**Table 3 ijerph-19-11539-t003:** The distribution of different *Leptospira* serovars found in each department included within the study.

Serovars	Total	Department				
		Antioquia	Cundinamarca	Valle del Cauca	Eje Afetero	Meta
Icterohaemorrhagiae	(21/77)	20/28	3/28	3/28	0/28	2/28
	27%	71.42%	10.7%	10.7%	0%	7.1%
Canicola	(12/77)	10/18	3/18	3/18	0/18	2/18
	16%	55.5%	16.6%	16.6%	0%	11.1%
Tarassovi	(9/77)	10/13	3/13	0/13	0/13	0/13
	12%	76.9%	23%	0%	0%	0%
Pomona	(9/77)	10/12	2/12	0/12	0/12	0/12
	11%	83.3%	16.6%	0%	0%	0%
Grippothyphosa	(8/77)	9/11	1/11	0/11	0/11	1/11
	10%	81.81%	9.09%	0%	0%	9.09%
Bratislava	(7/77)	7/9	0/9	1/9	1/9	0/9
	9%	77.7%	0%	11.1%	11.1%	0%
Ballum	(6/77)	7/8	0/8	0/8	0/8	1/8
	8%	87.5%	0%	0%	0%	12.5%
Autumnalis	(5/77)	6/7	0/7	1/7	0/7	0/7
	7%	85.71%	0%	14.28%	0%	0%

## Data Availability

The data presented in this study are available on request from the corresponding author. The data are not publicly available due to the privacy involved with the participants.

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
