# Peer review of "Seroprevalence and Molecular Characterization of Leptospira spp. in Rats Captured near Pig Farms in Colombia"

_ijerph, 2022, doi:10.3390/ijerph191811539_

Round 1
Reviewer 1 Report
My biggest issue with this manuscript is the use of nomenclature. I am not convinced if the authors know what they are measuring with the different methods. I will point this out as we go through the manuscript.
Is there any reason why no pigs were sampled in this study? This would greatly add value to this paper. Rodents are well known to carry Leptospira, however transmission routes are not so clear.
Line 20: Say PCR positivity instead of just positivity since positivity is defined by 2 tests in the paper.
Line 21: Delete serotypes. Pathogenicity is generally defined at species level. You can one serovar/serotype which will have pathogenic and saprophytic species
Line 59: Delete serovars. See comment above.
Line 104: MAT is used to identify Leptospira serovars, not species.
Line 107: From hereon, the authors refer to all MAT results as serovars of Leptospira interrogans whereas only serovars Pomona, Canicola, icterohaemorrhagiae, Bratislava and Autumnalis are from species L. interrogans. Ballum and Tarassovi are L. borgpeterseneii and Grippotyphosa is L. kirschneri.
Line 108: correct the spelling for Tarassovi
Line 131: Small S for seropositivity
Line 135: Is the 330 bp fragment from the 16s PCR different from the 331 bp fragment 16s PCR in line 143. There is no reference to either of the PCR.
Line 136: The reference for LipL32 PCR is to a 16s paper. The primers in this 16s is different from the primer sequence in line 145
Line 164-174: Group the species into pathogenic, intermediate and nonpathogenic so it is reader friendly.
Line 211: Replace interrogans with spp.
Table 2 title: Replace L. interrogans with Leptospira spp. Add in rodents to the title
Line 230: Change title to Distribution of Leptospira serovars in rodents by state.
Line 244: small S for seropositivity
Line 265: Check names for 40_LEP-F Consensus-RC and 254_LEP-F Consensus-RC. They do not match any names in the figure
Line 282: The present study identified 14% (IC95: 11.11-16:89) Leptospira seropositivity against 8 serovars in rodents captured on pig farms.
Line 316: prevalence of Leptospira, not leptospirosis. Leptospirosis is a disease, infection can be carried and detected by PCR without the disease.
Line 317: MAT only measures predetermined serovars of Leptospira while PCR can test for any serovar, however current methods can only identify this at species level.
Line 325: Use recent classification system as in https://doi.org/10.1371/journal.pntd.0007270
Line 355-357: Leptospira spp. 16S rRNA based phylogenetic analysis matched (47/50) samples with pathogenic Leptospira interrogans. This does not match the figure. In addition, line 249 says 22/500 samples positive to pathogenic Leptospira. This is in disagreement with 49 pathogenic samples (47 interrogans and 2 borgpeterseneii) in lines 356.
Author Response
Reviewer 1:
Comments
My biggest issue with this manuscript is the use of nomenclature. I am not convinced if the authors know what they are measuring with the different methods. I will point this out as we go through the manuscript.
Is there any reason why no pigs were sampled in this study? This would greatly add value to this paper. Rodents are well known to carry Leptospira, however transmission routes are not so clear.
Thank you very much for your comments and revisions; they were extremely helpful in improving the manuscript. Pigs could not be included in this study because the project's goal was to assess the presence and circulation of Leptospira in rats. We will take your important comment into account for future projects.
The text has been completely revised, and we hope that the combined corrections will clarify the inconsistencies found in the methods and results.
Line 20: Say PCR positivity instead of just positivity since positivity is defined by 2 tests in the paper. Done line 20
Line 21: Delete serotypes. Pathogenicity is generally defined at species level. You can one serovar/serotype which will have pathogenic and saprophytic species. DONE line 30
Line 59: Delete serovars. See comment above. DONE, line 63
Line 104: MAT is used to identify Leptospira serovars, not species. DONE, line 118
Line 107: From hereon, the authors refer to all MAT results as serovars of Leptospira interrogans whereas only serovars Pomona, Canicola, icterohaemorrhagiae, Bratislava and Autumnalis are from species L. interrogans. Ballum and Tarassovi are L. borgpeterseneii and Grippotyphosa is L. kirschneri. Done, the species were differentiated. Lines 122-124
Line 108: correct the spelling for Tarassovi= Done line 123
Line 131: Small S for seropositivity= done, line 154
Line 135: Is the 330 bp fragment from the 16s PCR different from the 331 bp fragment 16s PCR in line 143. There is no reference to either of the PCR= DONE. It was corrected to 330 bp It was a typo. Line 169
Line 136: The reference for LipL32 PCR is to a 16s paper. The primers in this 16s is different from the primer sequence in line 145. The adequate references were cited
Line 164-174: Group the species into pathogenic, intermediate and nonpathogenic so it is reader friendly. Done 200-221
Line 211: Replace interrogans with spp. DONE line 283
Table 2 title: Replace L. interrogans with Leptospira spp. Add in rodents to the title. Done see table 2
Line 230: Change title to Distribution of Leptospira serovars in rodents by state. Done table 3 title
Line 244: small S for seropositivity: done
Line 265: Check names for 40_LEP-F Consensus-RC and 254_LEP-F Consensus-RC. They do not match any names in the figure. Corrected
Line 282: The present study identified 14% (IC95: 11.11-16:89) Leptospira seropositivity against 8 serovars in rodents captured on pig farms. Done line 427-428
Line 316: prevalence of Leptospira, not leptospirosis. Leptospirosis is a disease, infection can be carried and detected by PCR without the disease. done. Line 472
Line 317: MAT only measures predetermined serovars of Leptospira while PCR can test for any serovar, however current methods can only identify this at species level. Done lines 474-476
Line 325: Use recent classification system as in https://doi.org/10.1371/journal.pntd.0007270. Done Lines 482-484
Line 355-357: Leptospira spp. 16S rRNA based phylogenetic analysis matched (47/50) samples with pathogenic Leptospira interrogans. This does not match the figure. Corrected lines 537-540
In addition, line 249 says 22/500 samples positive to pathogenic Leptospira. This is in disagreement with 49 pathogenic samples (47 interrogans and 2 borgpeterseneii) in lines 356. Done lines 537-539
Warm regards,
Jenny J Chaparro-Gutierrez

Reviewer 2 Report
Comments to: Seroprevalence and molecular characterization of Leptospira spp. in rodents captured near pig farms in Colombia. ijerph-1786546 – V.1
The paper describe a large study in Colombian pig-farms, which sought to assess the prevalence of Leptopira infection in rats collected at the farms. The participant collected 555 rats at 63 farms, which makes this a quite comprehensive study. Two methods were used for assessment of Leptospira infection: serology (MAT) and PCR on kidney tissues. Both methods can be considered as standard approaches for the assessment of Leptospira occurrence. The description of the rats, methods, calculations contain more detail than usual, and the presentation of the results and discussion hereof are generally adequate.
Major comments
Line 113: The participant have performed serological assays for multiple dilutions, and yet we are only informed on the prevalence in reference to the dilutions 1:100. This is quite common, but I think that it would be preferable and potentially more informative if we also get the prevalence rate for e.g. 1:200. Perhaps this will lead to alternative conclusions.
Line 246 – 249. Discusses the Leptospira types and reach the conclusion that 26% of the leptospira are saprophytic. This is based on the fact that they are positive in the RNA gene and negative in the LipL32. This deduction lead to the sensational and self-contradictory conclusion that the pathogenic leptospira are saprophytic.
I can’t figure out what went wrong – but the obvious conclusion is that the LipL32 did not work as intended, or that there was contamination of the PCRs for the RNA-PCR or both. Given the problems with the sequencing I would suspect the latter to be true. It can’t fully exclude that the findings are true since LipL32 is not required for infection (Murray, G. L., Srikram, A., Hoke, D. E., Wunder Jr, E. A., Henry, R., Lo, M., ... & Adler, B. (2009). Major surface protein LipL32 is not required for either acute or chronic infection with Leptospira interrogans. Infection and immunity, 77(3), 952-958.), but given that we with the term “saprophytic” also imply that other virulence factors are absent (Adler, B., & de la Peña Moctezuma, A. (2010). Leptospira and leptospirosis. Veterinary microbiology, 140(3-4), 287-296), then I must insist that the authors reassess and rethink the relevant paragraphs.
Minor comments
The title should refer to rats (Rattus sp.) since no other small animals are included.
There are several examples of imperfect sentences which must be corrected. It includes:
Line 16: rats not small animals (do this throughout the text)
Line 35: “insecurity”, which I assume refers to “food insecurity”.
Line 38: Earthquakes trigger disease outbreaks – which perhaps is true when the earth quakes leads to displacement of rodents or flooding, but there is no direct causative relationship.
Line 41: Colombia is theoretically vulnerable to high prevalence of leptospirosis – but I also think that our knowledge of animal farming at high altitudes are quite limited. I would suggest that the authors engage a bit more in explaining, why their results may be difficult to compare to other studies, due to differences in altitudes. This will also provide a convenient explanation for the lower prevalence at low altitude, as this just might be too warm (see: Jensen, P. M., & Magnussen, E. (2016). Is it too cold for Leptospira interrrogans transmission on the Faroese Islands?. Infectious Diseases, 48(2), 156-160.). Please be mindful of this when referring to other studies such as in line 285.
Line 48 – delete annually
Line 79: Please provide other terms than: “technified pig farm”.
Line 203-207. Belongs in discussion
Line 231: The sentence is convoluted – rewrite.
Line 235 – don’t write 0.000 – there must be a number at some point. Possibly just p< 0.0001
Line 250 to 256 belongs in M&M.
Line 311-312. The message is unclear.
There should be sufficient space for adding mean annual temperature, precipitation and altitudes for the five regions given in Table 2.
The print in Figure 1 is so small that I can’t read the text – a comment which also applies for Figure 2.
The manuscript will benefit from further corrections of similar nature, especially in table headings, which are not quite as helpful as they should be. Italics are missing in a few places: 357, 360..
In conclusion, a nice large study which need a more critical assessment – which could (or perhaps should) end in a deletion of the data from PCR-analyses. It will still be a nice paper if the authors expanded the serology a bit, and discussed the result in reference to the large gradient in altitude and temperatures.
Author Response
Dear Reviewer
We revised your suggestions, corrections, and comments one by one to improve the manuscript, and they were extremely helpful. Thank you so much.
Comments to: Seroprevalence and molecular characterization of Leptospira spp. in rodents captured near pig farms in Colombia. ijerph-1786546 – V.1
The paper describe a large study in Colombian pig-farms, which sought to assess the prevalence of Leptopira infection in rats collected at the farms. The participant collected 555 rats at 63 farms, which makes this a quite comprehensive study. Two methods were used for assessment of Leptospira infection: serology (MAT) and PCR on kidney tissues. Both methods can be considered as standard approaches for the assessment of Leptospira occurrence. The description of the rats, methods, calculations contain more detail than usual, and the presentation of the results and discussion hereof are generally adequate.
Reviewer 2
Major comments
Line 113: The participant have performed serological assays for multiple dilutions, and yet we are only informed on the prevalence in reference to the dilutions 1:100. This is quite common, but I think that it would be preferable and potentially more informative if we also get the prevalence rate for e.g. 1:200. Perhaps this will lead to alternative conclusions.
As the methodology said: Positive samples were placed in dilutions from 1:100 up to 1:3200 to determine antibody titers. Data not shown. We only obtain titer of 1:200 in 9/555 samples.
Line 246 – 249. Discusses the Leptospira types and reach the conclusion that 26% of the leptospira are saprophytic. This is based on the fact that they are positive in the RNA gene and negative in the LipL32. This deduction lead to the sensational and self-contradictory conclusion that the pathogenic leptospira are saprophytic.
Changed by: PCR results showed that 69% (344/500) of kidney samples were negative for Leptospira spp. and only 31.2% (156/500, IC95: 27.16-35.46) were positive. From these 26.8% (134/500 IC95:22.96-30.91) corresponded to Leptospira spp. (with only 16s RNA gene amplified) and 4% were positive for pathogenic Leptospira (22/500, IC95:2.78-6.59) (with LipL32 gene amplified). Lines 351-355
I can’t figure out what went wrong – but the obvious conclusion is that the LipL32 did not work as intended, or that there was contamination of the PCRs for the RNA-PCR or both. Given the problems with the sequencing I would suspect the latter to be true. It can’t fully exclude that the findings are true since LipL32 is not required for infection (Murray, G. L., Srikram, A., Hoke, D. E., Wunder Jr, E. A., Henry, R., Lo, M., ... & Adler, B. (2009). Major surface protein LipL32 is not required for either acute or chronic infection with Leptospira interrogans. Infection and immunity, 77(3), 952-958.), but given that we with the term “saprophytic” also imply that other virulence factors are absent (Adler, B., & de la Peña Moctezuma, A. (2010). Leptospira and leptospirosis. Veterinary microbiology, 140(3-4), 287-296), then I must insist that the authors reassess and rethink the relevant paragraphs.
Dear reviewer we have improved the description of methods, results, and discussion. We hope that the text could better now.
For this study, we performed two different PCRs.
One multiplex PCR based to detect 16S rRNA and gene LipL32 According to Ahmed et al. 2012. With this method we just checked the positivity for Leptospira sp. and the pathogenic Leptospira. LipL32 gene is a major outer-membrane lipoprotein from the genus Leptospira. Furthermore, the LipL32 gene is an important virulence factor confined to pathogenic strains of all Leptospira species. The mPCR uses 2 sets of primers, which are based upon amplification of the Leptospira 16S rRNA gene that is conserved throughout the bacterial kingdom, as well as another set of primers detecting the leptospiral major outer-membrane lipoprotein LipL32 gene. The LipL32 gene is highly specific and is absent from nonpathogenic leptospira or any other commensal pathogenic bacteria. The leptospiral major outer-membrane lipoprotein (LipL32) is expressed during infection by all pathogenic strains and can prove to be an important candidate antigen for the development of a sensitive and specific test for leptospirosis
The second PCR was conducted to amplify and then sequence a part of 16 sRNA. For this, we included only 50 samples from the positive ones,. The following primers were used: 5′-GGCGGCGCGTCTTAAACATG-3′ and 5′-GTCCGCCTACGCACCCTTTACG-3′; these primers have the ability to amplify all pathogenic and nonpathogenic leptospires (Djadid et al. 2009).
Minor comments
The title should refer to rats (Rattus sp.) since no other small animals are included. Done
There are several examples of imperfect sentences which must be corrected. It includes:
Line 16: rats not small animals (do this throughout the text)= We change rodent for Rats. Done in all text
Line 35: “insecurity”, which I assume refers to “food insecurity”= No, the term insecurity refers to precarious situation. Done the word was eliminated
Line 38: Earthquakes trigger disease outbreaks – which perhaps is true when the earth quakes leads to displacement of rodents or flooding, but there is no direct causative relationship. Done the word was eliminated
Line 41: Colombia is theoretically vulnerable to high prevalence of leptospirosis – but I also think that our knowledge of animal farming at high altitudes are quite limited. I would suggest that the authors engage a bit more in explaining, why their results may be difficult to compare to other studies, due to differences in altitudes. This will also provide a convenient explanation for the lower prevalence at low altitude, as this just might be too warm (see: Jensen, P. M., & Magnussen, E. (2016). Is it too cold for Leptospira interrrogans transmission on the Faroese Islands?. Infectious Diseases, 48(2), 156-160.). Please be mindful of this when referring to other studies such as in line 285. Table 2 now includes information on yearly mean temperatures, annual precipitation, and annual mean altitude. Additionally, we have discussed and drawn conclusions using this data.
Line 48 – delete annually Done
Line 79: Please provide other terms than: “technified pig farm”. done changed to Porkcolombia-associated
Line 203-207. Belongs in discussion= Done
Line 231: The sentence is convoluted – rewrite. Done
Line 235 – don’t write 0.000 – there must be a number at some point. Possibly just p< 0.0001. Done
Line 250 to 256 belongs in M&M.= done
Line 311-312. The message is unclear= done. We changed redaction
There should be sufficient space for adding mean annual temperature, precipitation and altitudes for the five regions given in Table 2. done
The print in Figure 1 is so small that I can’t read the text – a comment which also applies for Figure 2. Done
The manuscript will benefit from further corrections of similar nature, especially in table headings, which are not quite as helpful as they should be. Italics are missing in a few places: 357, 360. We have revised and corrected
In conclusion, a nice large study which need a more critical assessment – which could (or perhaps should) end in a deletion of the data from PCR-analyses. It will still be a nice paper if the authors expanded the serology a bit, and discussed the result in reference to the large gradient in altitude and temperatures.

Round 2
Reviewer 2 Report
Comments to: Seroprevalence and molecular characterization of Leptospira spp. rats captured near pig farms in Colombia. ijerph-1786546 – V.2
The authors submitted a revised version of the paper, which in general adequately addressed the comments given. I have a few more comments, pertaining to issues that I missed in the first review. The comments are as follows.
Major comments
Line 327: the prevalence rate is 14% given a 79 / 555. But the text, table 2, and table 3 only includes 77 positive animals – what happened to the last 2 positive animals?
Line 239: the prevalence rate was highest in Valle del Cauca (17%) – not in Cundinamarca (15%). It is correctly stated in line 311.
Figure 1. It a very nice map – but it does not bring any information, which is not already included in Table 2. I suggest you delete it.
Table 2. the last row has a total stated as “385”, which is quite confusing. The summation should state 13.8% (77/555) if you intend to summarize the above.
Line 260. It is meaningless to assess the number of positive R. norvegicus in reference to the total number of animals, since the latter number include other species. You may state that a disproportional number of infections were found R. norvegicus 63% (49/77).
Line 261 to 272: please write: there were significant difference between species as R. novegicus had twice the prevalence rate observed in r. rattus (Chi-sq. < 0.05: 8.83:CI 6.47-11.19 vs 5.41: CI 3.58-7.36). No differences was observed for juvenile vs. adults or for males vs. females (chi-sq., p’s > 0.05). You may hereafter feel free to delete table 4 and expand Figure 2 to a full page, which will allow the reader to assess the small print.
Table 4: probability of sero-positivity – surely not, since this would be 5 to 8%
Line 275: the word “only” is ill-used for a 31% prevalence rate.
Line 366: You need to provide some comment to the low number of Lipl32 positive. Perhaps – “some questions remain whether our Lipl32-PCR might have been negatively influenced, by e.g., inhibition.
Minor comments
Line 32. There should be a reference before the full stop. I suspect that ref no. 1 or 2 can be moved to this place instead of being listed in the following line.
Line 90-91. This line can, as a consequence of the revision, be deleted
Line 134. The prevalence calculation is used across many groups in the text, and the equation should reflect this. E.g. prevalence rate (subscript i,j) = number of positive (subscript i) / total examined (subscript j).
Line 140. Just Chi- square-test (no “d”)
Line 331: after periods: write: .., which could affect transmission, rat population dynamics, rat behavior etc. – add an “s” to association in the following line.
Line 343: I’m not convinced that the given studies can assess whether the rates get infected by the pigs, vice-versa or both. Reconsider the entire paragraph.
Line 369. I don’t understand why it is essential to characterize the Leptopira bacteria, unless you seek to assess transmission routes. Perhaps you can clarify that such insights may allow for improved control.
Table 3. Please add 77 before the word different.
References: ref. 28 and 41 is the same
References: ref. 29 and 42 is the same
Author Response
Dear Reviewer
Comments to: Seroprevalence and molecular characterization of Leptospira spp. rats captured near pig farms in Colombia. ijerph-1786546 – V.2
The authors submitted a revised version of the paper, which in general adequately addressed the comments given. I have a few more comments, pertaining to issues that I missed in the first review. The comments are as follows.
Major comments
Line 327: the prevalence rate is 14% given a 79 / 555. But the text, table 2, and table 3 only includes 77 positive animals – what happened to the last 2 positive animals?
Table 2 was deleted. We had a type error with the number of rats. There were only 77 seropositives (13,8%). This was corrected in the text.
Line 239: the prevalence rate was highest in Valle del Cauca (17%) – not in Cundinamarca (15%). It is correctly stated in line 311. Done
Figure 1. It a very nice map – but it does not bring any information, which is not already included in Table 2. I suggest you delete it. Done
Response: The point of the map was firstly and most importantly to show a map that would allow readers to visualize the regions in Colombia where the study took place and secondly (less important) to show the difference in seroprevalence by region. However, the review makes a valid point that the information their present is redundant and therefore the figure has been deleted.
Table 2. the last row has a total stated as “385”, which is quite confusing. The summation should state 13.8% (77/555) if you intend to summarize the above. Done
Line 260. It is meaningless to assess the number of positive R. norvegicus in reference to the total number of animals, since the latter number include other species. You may state that a disproportional number of infections were found R. norvegicus 63% (49/77).
We calculated the seroprevalence by species: The species Rattus norvegicus presented a seroprevalence of 11.9% (49/410, IC95: 8.8-15.0), while in Rattus rattus the seroprevalence was 20.7% ( 30/145, IC95: 14.1-27.3). line 376-380
Line 261 to 272: please write: there were significant difference between species as R. novegicus had twice the prevalence rate observed in r. rattus (Chi-sq. < 0.05: 8.83:CI 6.47-11.19 vs 5.41: CI 3.58-7.36). No differences were observed for juvenile vs. adults or for males vs. females (chi-sq., p’s > 0.05). You may hereafter feel free to delete table 4 and expand Figure 2 to a full page, which will allow the reader to assess the small print
DONE: we changed the sentence and deleted table 4.
Table 4: probability of sero-positivity – surely not, since this would be 5 to 8%. Table 4 was deleted
Line 275: the word “only” is ill-used for a 31% prevalence rate. Done
Table 4 was deleted
Line 366: You need to provide some comment to the low number of Lipl32 positive. Perhaps – “some questions remain whether our Lipl32-PCR might have been negatively influenced, by e.g., inhibition.
In this study, we used a PCR protocol that targets the lipL32 gene, which has high sequence conservation and is the most copious Leptospira protein (i.e. 38,000 copies per cell), thus supporting lipL32 as an appropriate target sequence. As we used mPCR for 16S rRNA gene and LipL32 gene, we can assure that there was no inhibition in the reaction. In this case the 16S rRNA function as housekeeping gene of the bacteria.
Minor comments
Line 32. There should be a reference before the full stop. I suspect that ref no. 1 or 2 can be moved to this place instead of being listed in the following line. Done
Line 90-91. This line can, as a consequence of the revision, be deleted. Done
Line 134. The prevalence calculation is used across many groups in the text, and the equation should reflect this. E.g. prevalence rate (subscript i,j) = number of positive (subscript i) / total examined (subscript j).
Done: it was added in all the document.
Line 140. Just Chi- square-test (no “d”): Done
Line 331: after periods: write: .., which could affect transmission, rat population dynamics, rat behavior etc. – add an “s” to association in the following line. Done lines 366-367
Line 343: I’m not convinced that the given studies can assess whether the rates get infected by the pigs, vice-versa or both. Reconsider the entire paragraph. Done lines 382-383
Line 369. I don’t understand why it is essential to characterize the Leptopira bacteria, unless you seek to assess transmission routes. Perhaps you can clarify that such insights may allow for improved control.
Thank for the comment. For us it is very important to characterize the Leptospira because there are few studies in our country about the bacteria in this particular group of rodents, that live near pig farm. We discuss this in lines 656 – 668.
Table 3. Please add 77 before the word different. Done
References: ref. 28 and 41 is the same. Corrected
References: ref. 29 and 42 is the same. Corrected
Murray GL. The lipoprotein LipL32, an enigma of leptospiral biology. Vet Microbiol. 2013;162(2–4):305–14.
